# Relationship between Motor Corticospinal System, Endogenous Pain Modulation Mechanisms and Clinical Symptoms in Patients with Knee Osteoarthritis: New Perspectives on an Old Disease

**DOI:** 10.3390/brainsci13081154

**Published:** 2023-08-01

**Authors:** Marylie Martel, Nathaly Gaudreault, René Pelletier, Francis Houde, Marie-Philippe Harvey, Caroline Giguère, Frédéric Balg, Guillaume Leonard

**Affiliations:** 1School of Rehabilitation, Faculty of Medicine and Health Sciences, Université de Sherbrooke, Sherbrooke, QC J1H 5N4, Canada; marylie.martel@usherbrooke.ca (M.M.); nathaly.gaudreault@usherbrooke.ca (N.G.); rene.pelletier.osteo@gmail.com (R.P.); 2Research Centre on Aging, CIUSSS de l’Estrie-CHUS, Sherbrooke, QC J1H 4C4, Canada; francis.houde@usherbrooke.ca (F.H.); marie.philippe.harvey@usherbrooke.ca (M.-P.H.); 3Centre de Recherche du Centre Hospitalier Universitaire de Sherbrooke, Sherbrooke, QC J1H 5N4, Canada; frederic.balg@usherbrooke.ca; 4Department of Diagnostic Radiology, Faculty of Medicine and Health Sciences, Université de Sherbrooke, Sherbrooke, QC J1H 5N4, Canada; caroline.s.giguere@usherbrooke.ca; 5Department of Surgery, Faculty of Medicine and Health Sciences, Université de Sherbrooke, Sherbrooke, QC J1H 5N4, Canada

**Keywords:** pain, knee osteoarthritis, transcranial magnetic stimulation, motor cortex, corticomotor excitability, motor evoked potentials, pain modulation, conditioned pain modulation, temporal summation

## Abstract

Knee osteoarthritis (OA) is a painful condition characterized by joint and bone changes. A growing number of studies suggest that these changes only partially explain the pain experienced by individuals with OA. The purpose of the current study was to evaluate if corticospinal and bulbospinal projection measurements were interrelated in patients with knee OA, and to explore the relationship between these neurophysiological measures and temporal summation (excitatory mechanisms of pain) on one hand, and clinical symptoms on the other. Twenty-eight (28) patients with knee OA were recruited. Corticospinal projections were measured using transcranial magnetic stimulation, while bulbospinal projections were evaluated with a conditioned pain modulation (CPM) protocol using a counter-irritation paradigm. Validated questionnaires were used to document clinical and psychological manifestations. All participants suffered from moderate to severe pain. There was a positive association between corticospinal excitability and the effectiveness of the CPM (r_s_ = 0.67, *p* = 0.01, *n* = 13). There was also a positive relationship between pain intensity and corticospinal excitability (r_s_ = 0.45, *p* = 0.03, *n* = 23), and between pain intensity and temporal summation (r_s_ = 0.58, *p* = 0.01, *n* = 18). The results of this study highlight some of the central nervous system changes that could be involved in knee OA and underline the importance of interindividual variability to better understand and explain the semiology and pathophysiology of knee OA.

## 1. Introduction

Osteoarthritis (OA) is the most common form of arthritis and remains one of the leading causes of pain in older adults, affecting more than 33% of people over the age of 65 [1]. Various treatments are available to alleviate OA symptoms and reduce its impact on physical function, such as therapeutic exercises, manual therapy, dry needling, and pain education [2,3]. Unfortunately, these approaches provide only partial relief for affected patients.

OA is traditionally viewed as a disease affecting the musculoskeletal system [4,5]. Nonetheless, recent data suggest that the central nervous system (CNS), via central sensitization and changes in the structure and function of the brain, can play an important role in OA pain [6,7,8]. Studies support the hypothesis that an impairment of endogenous pain control mechanisms, such as a deficit in pain-inhibiting mechanisms and/or an increase in pain-facilitating mechanisms, could play an important role in the development and maintenance of chronic pain [7,8,9]. The dysregulation of these pain mechanisms could explain, at least in part, poor treatment outcomes in some patients with chronic pain [9,10,11]. Petersen et al. demonstrated that patients with reduced pain inhibition combined with increased pain facilitation prior to knee arthroplasty were more likely to experience pain 12 months after surgery, compared to patients who exhibit only one of these impairments [9]. These lines of evidence challenged prior conceptions and prompted a redefinition of the physiopathology of OA, which is now considered as a multifactorial disease involving both musculoskeletal and neurophysiological mechanisms [7,12].

Conditioned pain modulation (CPM) and temporal summation are frequently used in research to assess pain inhibition and pain facilitation, respectively [13]. Decreased CPM and increased temporal summation are documented in many chronic pain conditions [10,11], including knee OA [9,14]. Interestingly, CPM alterations in OA patients tend to subside following successful knee arthroplasty [15]. This observation suggests that changes in bulbospinal projections could be considered a consequence of knee OA rather than a cause, thus supporting the idea that this pain-inhibiting mechanism may play an important role in pain modulation in this population.

A recent study found increased corticospinal excitability of the primary motor cortex (M1) to be associated with more efficient inhibitory pain modulation, as assessed by CPM, in healthy subjects [13]. However, it would be hasty to assume that this association is also present in pain populations among individuals with painful conditions such as OA. Thus, the objectives of the present study were to evaluate if corticospinal and bulbospinal projection measurements were interrelated in participants with knee OA and to explore the relationship between these neurophysiological measures and temporal summation (excitatory mechanisms of pain) on one hand, and clinical symptoms on the other.

## 2. Methods

### 2.1. Participants

Twenty-eight (28) persons with moderate-to-severe knee OA (mean age 69 ± 7 years old) agreed to participate in this study. For various reasons (described in the following sections), not all 28 participants completed all the experiments. To be included, participants were required to be 55 years old or over, have a diagnosis of knee OA by an orthopedist of the orthopedic clinic of the Centre intégré universitaire de santé et de services sociaux de l’Estrie–Centre hospitalier universitaire de Sherbrooke (CIUSSS de l’Estrie–CHUS), and have a referral for knee arthroplasty. Participants were asked to refrain from taking analgesics and consuming caffeine 6 h preceding each experimental session, and tobacco products 2 h prior to testing to avoid potential effects on pain and neurophysiological measures [16,17]. For security reasons, potential participants with neurological disorders, pacemakers, neurostimulators, metal implants, or epilepsy were excluded from the study. Individuals suffering from a painful condition (other than knee OA), or from any other major pathology, were also excluded. The research protocol was approved by the ethics committee of the CIUSSS de l’Estrie–CHUS (Sherbrooke, QC, Canada; approval # 2015-454-IUGS) and registered on ClinicalTrials.gov (#NCT03556423). Each participant provided informed written consent before participating in the study.

### 2.2. Inhibitory Pain Mechanisms (Bulbospinal Projections)

Conditioned pain modulation was assessed using a counter-irritation paradigm. First, pretests were used to familiarize participants with the Peltier-type thermode (30 × 30 mm, TSA-II, Medoc advanced Medical System, Israel) and the use of the computerized visual analog scale (CoVAS), and to determine the thermode temperature, for each participant, inducing a pain of 50/100 (moderate pain) [see [18] for more details]. Following the pretests, painful thermal stimuli were applied on the participant’s right forearm. The test stimulus was maintained at a constant temperature (determined during the pretest) for 2 min and the participants were asked to evaluate pain intensity with the CoVAS. Thereafter, the conditioning stimulus was performed by immersing the participant’s left forearm into cold water (10 °C) for 2 min, a procedure known as the cold pressor test (CPT). During the CPT procedure, pain intensity was evaluated every 30 s using a numerical pain rating scale (NRS, 0 = “no pain”, 100 = “worst imaginable pain”). Immediately after the CPT, the thermode was applied a second time using the same temperature. To facilitate comparisons for pain inhibitory mechanisms, pain intensity ratings obtained during the 2 min of the test stimulus were averaged and used in subsequent analyses. The magnitude of CPM was obtained by subtracting the post-CPT test stimulus pain scores from pre-CPT test stimulus pain scores, such that positive values represented an activation of inhibitory mechanisms.

### 2.3. Excitatory Pain Mechanisms (Temporal Summation)

The pain excitatory mechanisms (temporal summation) were measured using pain fluctuations, evaluated by the CoVAS during the 2 min when the test stimulus was applied prior to the conditioning stimulus, in the previous counter-irritation paradigm [18]. Because temporal summation can be quantified using several methods [13,18,19], we decided to calculate it by two different methods. First, temporal summation was calculated with the method used in Tousignant-Laflamme (2018), which proposes subtracting the pain score at 30 s from pain score at 120 s; a positive score corresponds to an increased activation of pain excitatory mechanisms. Second, temporal summation was calculated from the slope from a linear regression function obtained from the pain scores at 30, 60, 90, and 120 s. In this way, the risks of misattributing early, non-sustained fluctuations in pain scores are considerably reduced. Higher slope values represent an increased activation of excitatory mechanisms.

### 2.4. Corticomotor System (Corticospinal Projections and Intracortical Mechanisms)

Magnetic stimuli were delivered with a double-cone coil connected to a Magstim 200^2^ (mono-phasic single pulse; Magstim Co., Dyfed, UK). To record motor-evoked potentials (MEP), surface electromyography (EMG) electrodes (PicoEMG system; Cometa, Milan, Italy) were positioned over the *vastus lateralis* of the affected leg, according the SENIAM recommendations (http://www.seniam.org/; accessed on 28 July 2023). Electromyographic signals, elicited by magnetic stimuli, were amplified directly at the PicoEMG recording system (sampled at 2000 Hz, filtered at 1000 Hz, cutoff 20–2000 Hz) and then transferred to the acquisition computer using a Power 1401 mk II interface and Spike 2 software (version 7.10; Cambridge Electronic Design Limted, Cambridge, UK).

A neuronavigation system (Brainsight, Rogue Research Inc., Montreal, QC, Canada), was used to ensure the consistency of brain stimulation location over M1. To ensure optimal use of the neuronavigation system, a magnetic resonance imaging (MRI) session to obtain weighted brain anatomical images (T1; 1.0 mm × 1.0 mm × 1.0 mm, 3T Philips Medical system) was performed with each participant.

During the transcranial magnetic stimulation (TMS) procedure, the optimal location for eliciting MEP in the *vastus lateralis* (hotspot) was first identified. Subsequently, the TMS Motor Threshold Assessment Tool (MTAT 2.0), available online (http://www.clinicalresearcher.org/software.htm; accessed on 28 July 2023), was used to determine the resting motor threshold (rMT) for each participant, which was defined as the minimal intensity of stimulation required to induce MEP with an amplitude of over 50 μV. Then, with the participants at rest, eight blocks of different stimulation intensities (i.e., 90, 100, 105, 110, 120, 130, 140, and 150% of the rMT) were randomly administered, each block consisting of five magnetic stimuli (delay between each stimulation varied between 5 and 8 s). The peak-to-peak amplitude of MEP responses were measured offline and averaged for each patient to derive mean values. The slopes of the recruitment curves were calculated and used as a metric to reflect the strength of the corticospinal projections [20].

The cortical silent period (CSP) was measured to assess intra-cortical inhibition (Rossini, 2015). Five stimulations, performed at 120% of the rMT with the coil applied over the affected hemisphere, were delivered to participants while they actively contracted their quadriceps (10% of maximum voluntary contraction [MVC]; ~15 s between each stimulation). The CSP duration (time elapsing from onset of the MEP until the recurrence of voluntary tonic EMG activity) was calculated offline, trial-by-trial, and the individual mean value was used for subsequent analyses [21].

### 2.5. OA Signs and Symptoms Assessment

The Kellgren–Lawrence (KL) score, a radiographic classification reflecting the severity of articular damage, was determined by an experienced orthopedist (CG). Patients were also required to complete a series of questionnaires to assess pain and pain-related outcomes. Questionnaires were handed out at the first session, completed at home, and reviewed on arrival at the second session (see Figure 1). The selection of the different clinical measures used was based on the IMMPACT recommendations [22]. Specifically, participants completed two separate visual analog scales (VAS) to measure pain intensity and unpleasantness [23], the McGill Pain Questionnaire [MPQ] (qualitative aspect of pain) [24,25,26], the Brief Pain Inventory [BPI] (severity of pain and impact of pain on physical functioning) [27,28], the Western Ontario and McMaster Universities Osteoarthritis Index [WOMAC] (pain, stiffness, and functional mobility) [29,30,31], and the Central Sensitization Inventory [CSI] (central nervous system hypersensitivity) [32,33]; all these questionnaires were used to document OA clinical symptoms. The Pain Catastrophizing Scale [PCS] [34,35,36], the Tampa Scale of Kinesiophobia [TSK] [37,38,39], the Spielberger State-Trait Anxiety Inventory [STAI] [40,41], and the short form of the Beck Depression Inventory [BDI] [42,43,44] were also used to document psychological symptoms. Given the population studied, the validated French translations of all the questionnaires were used.

### 2.6. Statistical Analysis

Due to the small number of subjects and because visual inspection of the data (histograms) did not allow us to assume that the data were normally distributed, nonparametric tests were used. Spearman’s correlation analyses were performed to determine if corticospinal and bulbospinal measures were correlated, and to evaluate the relationship between these measures and clinical symptoms. Correlation coefficients were interpreted according to the classification of Mukaka: 0.9 to 1 = very highly correlated; 0.7 to 0.9 = highly correlated; 0.5 to 0.7 = moderately correlated; 0.3 to 0.5 = weakly correlated; and 0 to 0.3 = negligible [45]. For all analysis, statistical significance was set at *p* < 0.05, and statistical tests were performed using SPSS software (version 17; IBM Corp, Armonk, NY, USA).

## 3. Results

### 3.1. Participants’ Characteristics

From the 28 participants who were recruited, 5 did not complete the TMS evaluation (unable to find TMS rMT [*n* = 4], and unable to tolerate magnetic stimuli delivered by TMS [*n* = 1]). Additionally, MEP amplitudes at 150% of rMT could not be recorded in three other participants because the stimulation intensity was over the stimulator’s maximum output. Ten (10) participants were unable to complete the 2 min forearm immersion test in cold water (CPT), as the water temperature (10 degrees) caused intolerable pain. When participants were unable to keep their arm immersed for more than 60 s, their CPM data were excluded from the analyses. Every participant who completed the CPM procedure (*n* = 18) experienced the CPT as painful (all NRS ≥ 12, mean ± SD = 64 ± 21). The number of participants recruited is lower than originally planned (40) due to COVID-19. The general characteristics and the psychophysical, neurophysiological, and clinical measures of the participants are summarized in Table 1 and Table 2.

### 3.2. Correlation between CPM Responses and Corticomotor Excitability

Spearman analyses revealed the presence of a moderate and positive correlation between CPM responses and MEP amplitude at 110% of the rMT (see Figure 2), as well as a moderate and negative correlation between CPM and CSP (r_s_ = −0.69, *p* = 0.03, and *n* = 16). The slope of the recruitment curve was not correlated with CPM responses (*p* = 0.12).

### 3.3. Relationship between Inhibitory and Excitatory Pain Responses and Clinical Symptoms

Regarding the inhibitory pain responses, the results demonstrate a moderate correlation between CPM magnitude and the PCS (r_s_ = 0.59, *p* = 0.01), as well as between CPM magnitude and the STAI (state) questionnaire (r_s_ = 0.56, *p* = 0.03). No correlation was observed between CPM responses and the results from the scales and subscales of the MPQ (0.20 ≤ *p* ≤ 0.40), BPI (0.29 ≤ *p* ≤ 0.93), WOMAC (0.74 ≤ *p* ≤ 0.96), CSI (*p* = 0.10), TSK (*p* = 0.66), STAI-Trait (*p* = 0.10), BDI (*p* = 0.15), as well as for the VAS reflecting pain intensity (*p* = 0.23) and pain unpleasantness (*p* = 0.44).

Temporal summation was moderately and positively correlated with the pain section of the WOMAC. The correlations were the same for both of our metrics: delta score (r_s_ = 0.58, *p* = 0.01) and slope (r_s_ = 0.58, *p* = 0.01). No correlation was observed between temporal summation responses (for both metrics) and the results from the scales and subscales of the MPQ (0.32 ≤ *p* ≤ 0.99), BPI (0.13 ≤ *p* ≤ 0.76), WOMAC (0.25 ≤ *p* ≤ 0.33), CSI (0.28 ≤ *p* ≤ 0.41), PCS (0.11 ≤ *p* ≤ 0.29), TSK (0.56 ≤ *p* ≤ 0.66), STAI (0.66 ≤ *p* ≤ 0.91), BDI (*p* = 0.41), as well as for the VAS score, reflecting pain intensity (0.33 ≤ *p* ≤ 0.54) and pain unpleasantness (0.54 ≤ *p* ≤ 0.83).

### 3.4. Relationship between TMS Measures and Clinical Symptoms

The correlation coefficients depicting the relationship between TMS measures and clinical symptoms are presented in Table 3. As can be seen from this table, pain catastrophizing (PCS), minimal pain intensity (BPI), and the impact of pain on physical functioning (BPI) were the measures that appear to be the most consistently correlated with MEP amplitudes. No correlation was observed between the clinical symptoms and the rMT, the CSP, and the slopes of the recruitment curves. Interestingly, statistical analysis revealed a positive association between CSP and the KL score (r_s_ = 0.53, *p* = 0.03, and *n* = 16).

## 4. Discussion

The objectives of the present study were to assess whether bulbospinal and corticospinal projection measurements are interrelated in patients with knee OA, and to determine if there are relationships between these neurophysiological measures, temporal summation, and clinical symptoms of OA. Our results reveal that individuals with higher corticomotor excitability indices (MEP amplitudes at 110% of rMT) were those with higher CPM responses. We also observed a moderate and positive correlation between temporal summation and the pain section of the WOMAC, as well as between CPM responses and both pain catastrophizing and anxiety.

### 4.1. Relationships between Corticospinal and Bulbospinal Projections

Previous studies showed that stimulation of the motor cortex can bring relief to a number of patients with chronic pain [46]. Stimulation of M1 modulates corticospinal excitability and promotes the release of neurotransmitters (glutamate, acetylcholine, dopamine, and noradrenaline), which can in turn influence cortical and subcortical regions involved in pain perception and modulation [47,48,49,50,51], including the brainstem periaqueductal gray matter [52].

The results of Granovsky et al. allow us to further our understanding of the interaction that exists between pain and the motor system [13]. In their study, performed on healthy participants, the authors observed the presence of a moderate relationship between corticospinal excitability and CPM measures, suggesting possible links between corticospinal and bulbospinal descending projections [13]. The results obtained in our study confirm and extend the observations of Granovsky et al. by showing that increased corticospinal excitability measures are also associated with more efficient CPM responses in individuals suffering from knee OA pain.

### 4.2. Intracortical Inhibition

The cortical silent period is a TMS measure reflecting intracortical inhibition [21]. Contrary to MEP amplitude measures, which assess corticospinal pathways, CSP yields information about intracortical inhibitory phenomena [21]. The presence of pain can affect inhibitory (GABAergic) and excitatory (glutamatergic) neurotransmission and influence CSP duration [21,53]. Our results demonstrate that CSP measures were negatively correlated with CPM magnitude, indicating that individuals with longer intracortical inhibition also have lower CPM effects. These observations are contrary to those of Tarrago et al., who noted that patients with knee OA who showed greater intracortical disinhibition (*shorter* CSP duration) had lower CPM responses [54]. This discordance between our study and that of Tarrago et al. could be partly ascribed to the differences in TMS methodology. Indeed, Tarrago and collaborators measured the CSP over the right first dorsal interosseous (hand muscle) for all individuals, while we measured the CPS over the *vastus lateralis* (thigh muscle) of the affected limb.

### 4.3. Psychological Symptoms

People with chronic diseases are more likely to report psychological symptoms compared to people in good general health. Depression, pain catastrophizing, and anxiety are prevalent in people with knee OA [55,56]. Despite the fact that the participants in our study had mild psychological symptoms – with overall little heterogeneity – our results reveal that those with higher levels of pain catastrophizing and anxiety had higher CPM responses. To date, the scientific literature is divergent regarding the relationship between psychological symptoms and CPM responses. While a number of studies failed to find an association between levels of pain catastrophizing and CPM [57,58], others observed a positive [59] or negative [60] association between these two variables. The study by Nahman-Averbuch et al. helps to shed some light on these contradictory results by showing that the relationship between these variables would be specific to the pain modality used to evaluate CPM [61], suggesting that these distinct CPM assessment methods reflect different neurophysiological mechanisms [62].

### 4.4. Temporal Summation

Previous studies showed that patients suffering from knee OA tend to show increased temporal summation responses, compared to healthy (pain-free) individuals [9,14,63]. While these observations are interesting, they do not reflect the inter-individual variations of temporal summation responses in the population with knee OA. Our results extend these observations by showing that the magnitude of the temporal summation varies between individuals with knee OA and that these variations are positively correlated with WOMAC pain measures. These results are consistent with the results of Kurien et al. and of Neogi et al., who measured pain symptoms using the PainDETECT and the WOMAC, respectively [63,64].

Interestingly, Petersen and colleagues observed that patients who had higher levels of pain 12 months after total knee replacement were those who had higher temporal summation responses before the surgery, suggesting that temporal summation may be a good predictor of persistent pain [65]. In keeping with this idea, Yartnisky and colleagues proposed a clinical spectrum, the “pain modulation profile”, where each individual can be positioned between pronociceptive and antinociceptive, according to his/her temporal summation and CPM responses [66]. Individuals with higher temporal summation and/or lower CPM responses (antinociception profile) would hence be more likely to develop chronic pain. For patients with knee OA pain, this clinical spectrum could potentially help better identify the individuals most likely to benefit from knee arthroplasty, echoing the idea of personalized medicine that would tailor treatments to patients’ “pain modulation profile”.

### 4.5. Clinical and Rehabilitation Perspectives

OA studies looking into the relationship between corticomotor excitability, pain, and disability show divergent results [54,67,68]. In this study, patients reporting higher levels of pain and disability tended to show higher MEP responses. The same was true for pain catastrophizing, which is perhaps less surprising, considering what we know about this psychological trait and its association with pain intensity and disability [55,69,70]. Taken together, the results of the present study re-emphasize the potential central role of the cortico-motor system in knee OA, highlighting not only the links between the motor system and patient-reported symptoms (corticospinal excitability, pain, and disability), but also those between cortico-cortical mechanisms (CSP) and joint alterations (KL radiographic score), once again underlining the potential contribution of M1 maladaptive plasticity in chronic pain [71,72,73].

### 4.6. Limits

This study has some limitations. First, although 28 patients were initially recruited, less than half completed both TMS and CPM evaluation. These contingencies, combined with the small number of participants recruited (smaller than initially anticipated, due to the COVID pandemic), decrease statistical power and increase the risk of type II errors. It is therefore possible that other associations could not be identified, reminding us that non-significant correlations should be interpreted with caution. Other limitations concern the fact that TMS experimentations were performed only on the quadricep muscle (*vastus lateralis*) of the affected knee. Additional TMS measure (for instance, of a distant muscle or of the contralateral “healthy” quadriceps muscle) would help to better circumscribe the effects and determine whether the observed changes are widespread or localized to the affected OA joint. However, adding other TMS measures would require much more time, which we preferred to avoid, considering the already lengthy testing session.

We must also bear in mind that the present results apply only to the population studied. At this stage, it might be interesting to determine whether these observations are similar for all OA populations, or whether they are specific to knee OA patients. Although great care must be taken to avoid erroneous generalizations, the results of Sánchez-Romero et al. [74], who showed no difference between hip and knee OA patients for a series of hematological markers, could suggest that the associations observed in this study could also extend to other OA populations. Further studies, evaluating different OA patient populations, are obviously required before any conclusions can be drawn.

## 5. Conclusions

The current results reveal that there is a positive correlation in knee OA patients between pain intensity and corticospinal excitability, as well as between pain intensity and temporal summation. An association between corticospinal excitability and CPM efficacy was also observed, indicating that patients with higher corticomotor excitability tend to have higher CPM responses. These observations support the need for further studies looking into interindividual variability to better understand OA pain symptoms and apprehend the involvement of the CNS in the pathophysiology of knee OA.

## Figures and Tables

**Figure 1 brainsci-13-01154-f001:**
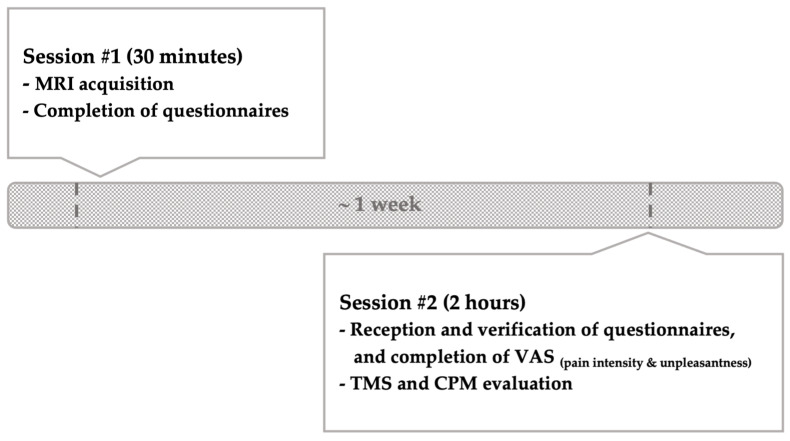
Timeline of the study.

**Figure 2 brainsci-13-01154-f002:**
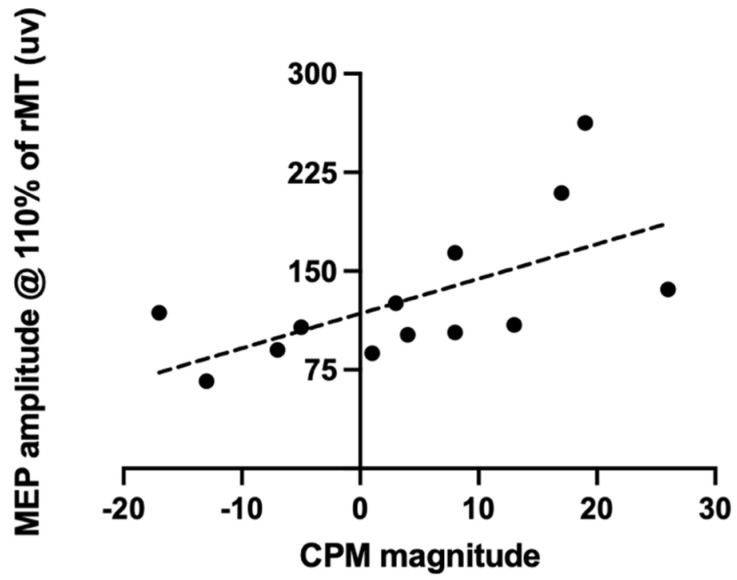
CPM responses were positively correlated with the MEP amplitude at 110% of the rMT (r_s_ = 0.67, *p* = 0.01, and *n* = 13).

**Table 1 brainsci-13-01154-t001:** Participants’ characteristics.

Variable	
**Number (*n*)**	28
**Gender (F/M)**	14/14
**Age (years)**	
**Mean ± SD**	69 ± 7
**Range**	57–82
**Time since diagnosis (years)**	
**Mean ± SD**	5.5 ± 5.5
**Kellgren-Lawrence score (%)**	
**KL 2**	4
**KL 3**	18
**KL 4**	78

SD: standard deviation.

**Table 2 brainsci-13-01154-t002:** Psychophysical, neurophysiological, and clinical measures.

Psychophysical Variables	Mean	Standard Deviation	*n*
**CPM**			
*Mean pain before CPT ( /100)*	49.9	17.9	18
*Mean pain after CPT ( /100)*	47.0	17.5	
*Delta score*	2.9	12.7	
**Temporal summation**			
*120–30 s*	14	25	18
*Slopes*	19	25	
**Neurophysiological variables**			
**rMT** *(% of the stimulator’s maximum output)*	51	8	23
**MEP amplitude (µV) @**			
*100% of rMT*	127.60	69.47	16
*105% of rMT*	129.53	62.45	17
*110% of rMT*	135.57	71.26	23
*120% of rMT*	243.18	134.87	17
*130% of rMT*	243.81	150.95	22
*140% of rMT*	370.91	240.28	16
*150% of rMT*	377.00	256.92	20
**Slope of the recruitment curves**	5.38	4.91	22
**CSP (ms)**	183.4	48.2	16
**Clinical symptoms**			
**MPQ**			28
*Global score ( /72)*	21	13	
*Numerical rating scale ( /10)*	4	3	
**BPI**			28
*Worst pain last 24 h ( /10)*	6	2	
*Least pain last 24 h ( /10)*	3	2	
*Pain on average ( /10)*	5	2	
*Pain right now ( /10)*	4	3	
*Incapacity ( /70)*	29	14	
**WOMAC**			28
*Pain ( /500)*	254	110	
*Stiffness ( /200)*	117	50	
*Functional mobility ( /1 700)*	860	369	
**CSI ( /100)**	31	15	22
**PCS ( /52)**	14	12	28
**TSK ( /68)**	39	9	22
**STAI**			22
*State ( /80)*	33	11	
*Trait ( /80)*	35	10	
**BDI ( /39)**	3	3	28
**Pain unpleasantness (VAS)**	3	3	22

Abbreviations: rMT; resting motor threshold, MEP; motor evoked potential, CSP; cortical silent period, CPM; conditioned pain modulation, MPQ; McGill Pain Questionnaire, BPI; Brief Pain Inventory, WOMAC; Western Ontario and McMaster Universities Osteoarthritis Index, CSI; Central Sensitization Inventory, PCS; Pain Catastrophizing Scale, TSK; Tampa Scale of Kinesiophobia, STAI; Spielberger State-Trait Anxiety Inventory, BDI; Beck Depression Inventory, and VAS; visual analog scale.

**Table 3 brainsci-13-01154-t003:** Correlations between MEP amplitudes at different TMS intensities and clinical questionnaires scores (*n* = 23).

TMS Intensity (% of rMT)	110	120
**MPQ**		
*Global score*	NS	NS
*Numerical rating scale*	rs = 0.50, *p* = 0.02	rs = 0.65, *p* < 0.01
**BPI**		
*Pain at its worst (last 24 h)*	rs = 0.54, *p* < 0.01	NS
*Pain at its least (last 24 h)*	rs = 0.44, *p* = 0.04	rs = 0.50, *p* = 0.04
*Pain interference (last 24 h)*	rs = 0.49, *p* = 0.02	NS
*Pain on average*	rs = 0.55, *p* < 0.01	NS
*Present pain*	rs = 0.51, *p* = 0.01	rs = 0.64, *p* < 0.01
**WOMAC**		NS
*Pain*	rs = 0.45, *p* = 0.03	
*Stiffness*	NS	
*Functional mobility*	NS	
**CSI**	r_s =_ 0.63, *p* < 0.01	NS
**PCS**	r_s =_ 0.43, *p =* 0.04	r_s =_ 0.50, *p =* 0.04
**TSK**	NS	NS
**STAI**	NS	NS
*State*		
*Trai*		
**BDI (short form)**	NS	NS
**Pain intensity** *****	NS	r_s =_ 0.53, *p =* 0.03
**Pain unpleasantness** *****	NS	NS

Abbreviation: MEP; motor evoked potential, TMS; transcranial magnetic stimulation, rMT; resting motor threshold, MPQ; McGill Pain Questionnaire, BPI; Brief Pain Inventory, WOMAC; Western Ontario and McMaster Universities Osteoarthritis Index, CSI; Central Sensitization Inventory, PCS; Pain Catastrophizing Scale, TSK; Tampa Scale of Kinesiophobia, STAI; Spielberger State-Trait Anxiety Inventory, BDI; Beck Depression Inventory, VAS; visual analog scale, NS; non-significant, and * measured with VAS.

## Data Availability

Not applicable.

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
