# Peer review of "Relationship between Motor Corticospinal System, Endogenous Pain Modulation Mechanisms and Clinical Symptoms in Patients with Knee Osteoarthritis: New Perspectives on an Old Disease"

_brainsci, 2023, doi:10.3390/brainsci13081154_

Round 1
Reviewer 1 Report
1-"Different questionnaires were used to document the clin- 25 ical and psychological manifestations" Please specify.
2-"There 26 was a positive association between corticospinal excitability and the effectiveness of the CPM. There 27 was also a positive relationship between pain intensity, corticospinal excitability, and temporal 28 summation." If this is a result of a correlation analysis, please provide the wieght of the corelation along with the correlation coefficients.
3-Why did you set the age limit as 55 years?
4-"have a diagnosis of knee OA" according to..?
5-Did you exclude patients with major organ dysfunction? Exclusion criteria requires some more detail.
6-Please provide country of origi for all devices and software.
7-"Spearman correlation analyses" Spearman's please...
8-How was the sample size arrived at?
9-". Ten (10) participants were unable to support the CPT and where 185 therefore unable to complete the CPM procedure." What was the reason?
10-"Kellgren-Lawrence score Mean ± SD 3.8 ± ..." How reliable is it to provide mean score for KL; it is a categorical variable.
11-Table 1: Is it correct to provide the sample size as 28? There are patients who could not complete the test(s).
12-Spearman's rho should not be presented as "r".
13-What was the role of the funder?
14-Did each author meet the 1st criterion of ICMJE4 authorship criteria?
15-The study could be supported by figures.
Reviewer 2 Report
The authors have produced an interesting and useful study to evaluate if corticospinal and bulbospinal projection measurements were interrelated in patients with knee OA, and to explore the relationship between these neurophysiological measures and temporal summation (excitatory mechanisms of pain), on one hand, and clinical symptoms, on the other.
However, I would like to make some observations before recommending your work for publication.
1. Is the study a secondary analysis of a clinical trial? If so, please detail it in the title and in the methods section.
2. In the introduction, I recommend that the authors to comment on the conservative treatment that can be very useful in mild to moderate OA. I recommend using and mentioning the following quality papers of studies investigating the use of Dry Needling and a 3-month program of Therapeutic Exercise, Manual Therapy, Therapeutic Exercise, and Pain Education: doi:10.1093/pm/pnz036, doi:10.3390/app11041895,
3. Could the authors add a Graphical Abstract?
4. There is an interesting work that develops to identify the differences in blood investigations between total hip and total knee replacement. I recommend the authors to discuss it: 10.1097/TGR.0000000000000337
5. I recommend summarizing the conclusions section a bit, describing it by essential points that respond to the objectives.
Reviewer 3 Report
Dear Authors,
The purpose of the current study was to evaluate if corticospinal and bulbospinal projection measurements were interrelated in patients with knee OA, and to explore the relationship between these neurophysiological measures and temporal summation (excitatory mechanisms of pain), on one hand, and clinical symptoms, on the other. This is an important study. The manuscript is well written. However, some improvements are needed to improve its readability and reliability.
My main concern is the research methodology. It should be presented in more detail.
The recruiting the subjects in the study should be described in more detail. What was the total number of subjects examined? What is the From paragraph 3.1 this can be seen as 18. Then table 1 presents incorrect information. How the required number of subjects was calculated? What was the power of the study?
In the abstract, the results should be presented with estimated parameters.
Reviewer 4 Report
Dear Authors
I was pleased to read the paper entitled "Relationship between Motor Corticospinal System, Pain Mechanisms, and Clinical Symptoms in Patients with Knee Osteoarthritis: A Cross-Sectional Study"
The study appears well-described, clear, and concise.
The nature of cross-sectional study is not immediately understandable; The part about methods is written very well, but if you want to underline this aspect already in the title, it should be better clarified with a graph or other way to highlight the simultaneous reading of the exposure and the outcome. At first glance, it looks like an observational study in which a diagnostic method is applied.
In method line 86, I would remove the extended definition with the acronym "Conditioned pain modulation (CPM)" already defined in the introduction.
Figure 1, relating to the timeline, is represented with an arrow which suggests that there is something even after session 2.
Maybe just my interpretation and impression, but think about whether to change the representation
For the rest, I have no particular notes relating to a well-executed and reported study. Even the limits you mentioned are correct and identify the aspects to deepen new studies (particularly for the study of muscles not correlated to OA pathology).
Round 2
Reviewer 2 Report
The authors have responded to my suggestions and questions, and the current version of their manuscript is of high quality, so I recommend its publication.
Congratulations
Author Response
We thank the reviewer for her/his thoughtful suggestions. We believe that the revision has significantly improved our manuscript.
Reviewer 3 Report
Dear Authors,
The objectives of the present study were to determine if bulbospinal and corticospinal projection measurements were interrelated in participants with knee osteoarthritis (OA) and to explore the relationship between these neurophysiological measures on the one hand, and temporal summation and OA clinical symptoms on the other.
Several improvements have been made. However, the manuscript still has flaws in methodology. The statistical analysis plan must be clearly presented. Since the main results were obtained with a total of 18, please provide the characteristics of the subjects who participated in all measurements.
The recruiting of study group, inclusion and exclusion criteria as well subjects characteristics are best presented in the section "Methodology". Some information about the health status of the subjects is also appreciated.
Please check abbreviations and indexing. It seems that not everything is explained.
The Abstract should more clearly represent the study. Please, edit the study purpose to be identical to the maintext. The study methodology and results should be presented. The total number (n=18) should be noted.
